# Structural mechanism of intracellular autoregulation of zinc uptake in ZIP transporters

Changxu Pang [1,3], Jin Chai [1,3], Ping Zhu [1], John Shanklin [1] & Qun Liu [1,2] ✉

Zinc is an essential micronutrient that supports all living organisms through regulating numerous biological processes. However, the mechanism of uptake regulation by intracellular $Zn^{2+}$ status remains unclear. Here we report a cryo-electron microscopy structure of a ZIP-family transporter from *Bordetella bronchiseptica* at 3.05 Å resolution in an inward-facing, inhibited conformation. The transporter forms a homodimer, each protomer containing nine transmembrane helices and three metal ions. Two metal ions form a binuclear pore structure, and the third ion is located at an egress site facing the cytoplasm. The egress site is covered by a loop, and two histidine residues on the loop interact with the egress-site ion and regulate its release. Cell-based $Zn^{2+}$ uptake and cell growth viability assays reveal a negative regulation of $Zn^{2+}$ uptake through sensing intracellular $Zn^{2+}$ status using a built-in sensor. These structural and biochemical analyses provide mechanistic insight into the autoregulation of zinc uptake across membranes.

Zinc is an earth-abundant transition-metal micronutrient. Carrying two positive charges, $Zn^{2+}$ interacts with numerous biological molecules including proteins and nucleotides. It is estimated that ~10% of eukaryotic proteins and 5–6% of prokaryotic proteins contain $Zn^{2+}$[1]. Hence, $Zn^{2+}$ is essential for physiological functions in gene transcription, protein structural stability, development, photosynthesis, immune response, and many enzyme activities[2]. Nevertheless, too much $Zn^{2+}$ is toxic and its cellular concentration must be tightly regulated within the nanomolar range.

For $Zn^{2+}$ to perform its biological roles, it must be first taken up from the environment through zinc transporters. Three main types of $Zn^{2+}$ uptake transporters have been identified: Zrt-/Irt-like proteins (ZIPs)[3,4], ATP-binding cassette (ABC) transporters[5], and natural resistance-associated macrophage proteins (NRAMP)[6]. In addition to $Zn^{2+}$ uptake transporters, $Zn^{2+}$ efflux transporters such as ZnTs (zinc transporters) and ATPases export intracellular $Zn^{2+}$ to prevent zinc cytotoxicity[2,5]. The synergic actions of diverse uptake and export transporters contribute to the regulation of $Zn^{2+}$ homeostasis within a cell and the whole organism. Disruption of $Zn^{2+}$ homeostasis, resulting in either $Zn^{2+}$ deficiency or excess may cause metabolic disruption, disease, or ultimately death of the organism[3,7].

Due to their key role in $Zn^{2+}$ homeostasis, ZIP transporters are broadly represented in animals, plants, protozoans, prokaryotes, and archaea[8]. In humans, ZIPs belong to the solute carrier 39 (SLC39) family which consists of 14 members[9]. Human ZIPs are named ZIP1 to ZIP14 and are localized to different tissues and cellular membranes for zinc uptake and mobilization to various subcellular organelles. The malfunction and genetic mutations of specific human ZIPs have been linked to diseases including ZIP4 in acrodermatitis enteropathica (AE)[10], ZIP6 in esophageal squamous cell carcinoma (ESCC)[11], ZIP10 in skin diseases[12], ZIP12 in pulmonary hypertension[13], and ZIP14 in cancer cachexia muscular atrophy[14]. In the plant *Arabidopsis thaliana*, there are 15 ZIPs[4]. Like their human relatives, plant ZIPs are responsible for the uptake of $Zn^{2+}$ in this case from the soil by root cells, mobilizing it to plant cells as well as intracellular organelles. In rice, *Os*ZIP5 and *Os*ZIP9 are reported to uptake $Zn^{2+}$ under $Zn^{2+}$ limiting conditions[15–17]. Plant ZIP members are transcriptionally regulated in response to $Zn^{2+}$-deficient and toxic soils[4,18].

[1]Biology Department, Brookhaven National Laboratory, Upton, NY, USA. [2]NSLS-II, Brookhaven National Laboratory, Upton, NY, USA. [3]These authors contributed equally: Changxu Pang, Jin Chai. ✉e-mail: qunliu@bnl.gov

In bacteria, there are two main zinc uptake transporters, the ZnuABC transporter and the ZIP family transporter[19]. $Zn^{2+}$ transporters in pathogenic bacteria play essential roles in the processes of infection and pathogenicity in response to "nutritional immunity" in which the host immune system prevents zinc from being utilized by pathogens[20]. ZupT proteins are members of the ZIP family[21]. In several pathogens, for example, *Salmonella enterica*[22,23], *Clostridioides difficile*[24], and pathogenic *Escherichia coli*[25], ZupT proteins contribute to infection and pathogenicity. Modulating zinc uptake in pathogenic bacteria may provide a strategy for the treatment of infectious diseases.

To understand the function and mechanism of how ZIP transporters work, the structure of a bacterial ZIP transporter from *Bordetella bronchiseptica* (*Bb*ZIP) was determined by X-ray crystallography[26]. The structure was determined as a monomer in an inward-facing conformation. Conserved residues in a transmembrane pore defined binuclear sites (M1 and M2) which may be occupied by $Zn^{2+}$ or $Cd^{2+}$. To understand the transport mechanism, a two-domain elevator model of the *Bb*ZIP dimer was proposed based on the coevolutionary network of interface residues[27]. *Bb*ZIP has a molecular weight of 31 kDa. However, its crystal structure contains only 224 ordered residues with a molecular weight of 22.1 kDa[26]. Nearly 30% of the *Bb*ZIP structure is missing in the crystal structure. *Bb*ZIP has been reported as a dimer[28]. However, the lack of a *Bb*ZIP dimer structure has prevented further understanding of the current model.

ZIP transporters are tightly regulated translationally and post-translationally[19,29]. For human ZIP4, intracellular $Zn^{2+}$ levels regulate ZIP function through endocytosis; elevated $Zn^{2+}$ triggers endocytosis of ZIP4, and a cytoplasmic histidine-rich domain is targeted for ubiquitination that leads to its degradation[30]. A site involved in $Zn^{2+}$ transport in ZIP4 was suggested to sense intracellular $Zn^{2+}$ status to regulate endocytosis[31]. In plant *Arabidopsis thaliana* IRT1, a ZIP member, a histidine-rich intracellular loop between TM3-4 was suggested to sense $Zn^{2+}$. Consistent with this view, truncating the loop resulted in the accumulation of $Zn^{2+}$. It was shown that the binding of $Zn^{2+}$ to the loop promoted ubiquitination, endocytosis, and vacuolar degradation[32]. Bacterial ZIPs do not have a large intracellular domain, but contain a histidine-rich loop between TM3-4. Little is known about the function of the loop in regulating $Zn^{2+}$ uptake. We thus hypothesized that the loop might sense intracellular $Zn^{2+}$ status as part of a mechanism to regulate its own transport activity.

In this work, we determined a dimer structure of *Bb*ZIP in an inward-facing, inhibited conformation. An intracellular histidine-rich loop between TM3-4 folds back to the inward-facing pocket and we present evidence that it is part of an intracellular $Zn^{2+}$ sensor that regulates $Zn^{2+}$ uptake. Cell-based $Zn^{2+}$ uptake and cell-growth viability assays are consistent with the two histidines on the loop being involved in sensing and autoregulating $Zn^{2+}$ uptake and intracellular $Zn^{2+}$ homeostasis. Our structure-function characterization of *Bb*ZIP reveals a potentially conserved mechanism of regulation of $Zn^{2+}$ uptake through a built-in intracellular $Zn^{2+}$ sensor.

## Results

### Structure determination

Cryo-electron microscopy (cryo-EM) structure analysis of small membrane proteins remains challenging, in particular for membrane proteins without an external domain that serves as an important feature in the alignment of particles[33]. To facilitate cryo-EM analysis of small membrane proteins, antibodies and nanobodies are commonly used to increase the molecular size and serve as alignment markers[34]. However, developing high-affinity antibodies and nanobodies takes tremendous amounts of effort and time. In our SDS-PAGE gel and size-exclusion chromatography analyses, the *Bb*ZIP dimer has an estimated molecular weight of only about 50–60 kDa (Supplementary Fig. 1a, b). In addition, the crystal structure of *Bb*ZIP shows that it doesn't have an

extracellular or intracellular domain, making a challenging case of structural analysis by cryo-EM.

To overcome the non-outer membrane issue, we reconstituted purified *Bb*ZIP into the small amphipol PMAL-C8 to form *Bb*ZIP-PMAL-C8 nanoparticles. For small membrane proteins of about 60 kDa and particle sizes of 60–70 Å in diameter, collecting data at a high magnification would provide enough pixels to more accurately estimate the noise of both background and particle signals. To enhance the signal-to-noise ratio of *Bb*ZIP images, we used a physical pixel size of 0.666 Å/pixel in super-resolution mode, used an energy filter width at 15 eV, and collected cryo-EM data from *Bb*ZIP particles embedded in thin ice. On collected micrographs, *Bb*ZIP particles have dimensions of roughly 50 – 100 Å with no apparent aggregates (Supplementary Fig. 1c). 2D class averaging of selected particles showed multiple views of dimer features with 2-fold symmetry (Supplementary Fig. 1d). Transmembrane helices are seen as rods from side views and spots from top views, indicating the high quality of the prepared particles and collected data (Supplementary Fig. 1d). Amphipol polymers can be seen as rectangular belts when viewed from the top.

Due to the presence of amphipols around *Bb*ZIP, standard ab initio 3D reconstruction in cryoSPARC[35] did not produce reasonable structural models. To make use of the structural features of transmembrane helices in ab initio reconstruction, we used data with a high-resolution cutoff at 7 Å to generate four ab initio models from 50,000 randomly selected particles belonging to 135 classes (Supplementary Fig. 2) in cryoSPARC. Subsequently, the four classes were used for iterative 3D heterogenous refinements first at 64 pixels and then at 128 pixels to obtain classes with the best α-helical features. Supplementary Fig. 3 summarizes the workflow that we used to analyze the cryo-EM data for the reconstruction of *Bb*ZIP. We think that the workflow may also work for other small membrane proteins.

We determined the structure in the presence of amphipols. Although without any structural features, amphipol densities contribute to the reconstruction as indicated by the three views of the reconstructed map colored by local resolutions (Supplementary Fig. 4a–c). Amphipols wrapped *Bb*ZIP tightly with almost no extra protein densities outside, indicating a compact *Bb*ZIP structure embedded in the amphipol nanodisc (Supplementary Fig. 4a). The final refined dimer reconstruction contains 37,954 particles at a resolution of 3.05 Å as estimated by the gold standard Fourier Shell Correlation (FSC) at 0.143 (Supplementary Fig. 4d). In the final reconstruction, most particles represent a wide-spread distribution in the Euler space with slightly more side-view particles (Supplementary Fig. 4e). The reconstructed map used 2-fold symmetry (C2) which displays a higher resolution (3.05 Å) than a separate reconstruction with no symmetry (C1) (3.3 Å) (Supplementary Fig. 4d). We checked the two protomers in the C1 reconstruction and we did not observe structural differences.

### Dimer structure

The reconstructed *Bb*ZIP dimer is of high quality (Fig. 1a). Transmembrane helices are resolved with characteristic side chains to assist model building and refinement (Supplementary Fig. 5). The dimer has its N-termini in the cytosol and C-termini in the periplasm with the 2-fold symmetry (Fig. 1a–c). The dimer structure consists of 18 transmembrane helices, nine from each protomer. The N-terminus has an amphipathic α-helix Ha which is roughly perpendicular to TMb (Fig. 1b, c). $Cd^{2+}$ has been shown to help stabilize the protein for crystallization[26], so we used $Cd^{2+}$ in our protein purification and found six $Cd^{2+}$ ions in the dimer. Furthermore, we identified lipid densities that may be best modeled as phosphatidylethanolamine (PE), which is the most abundant lipid in *Escherichia coli*. Two modeled PE molecules interact with both monomers and might contribute to the stabilization of the dimer (Fig. 1c). The dimer interface is primarily formed by TM3, TM7, and TM8 from each monomer with an area of 1688 Å$^2$ as calculated by the program PISA[36]. The formation

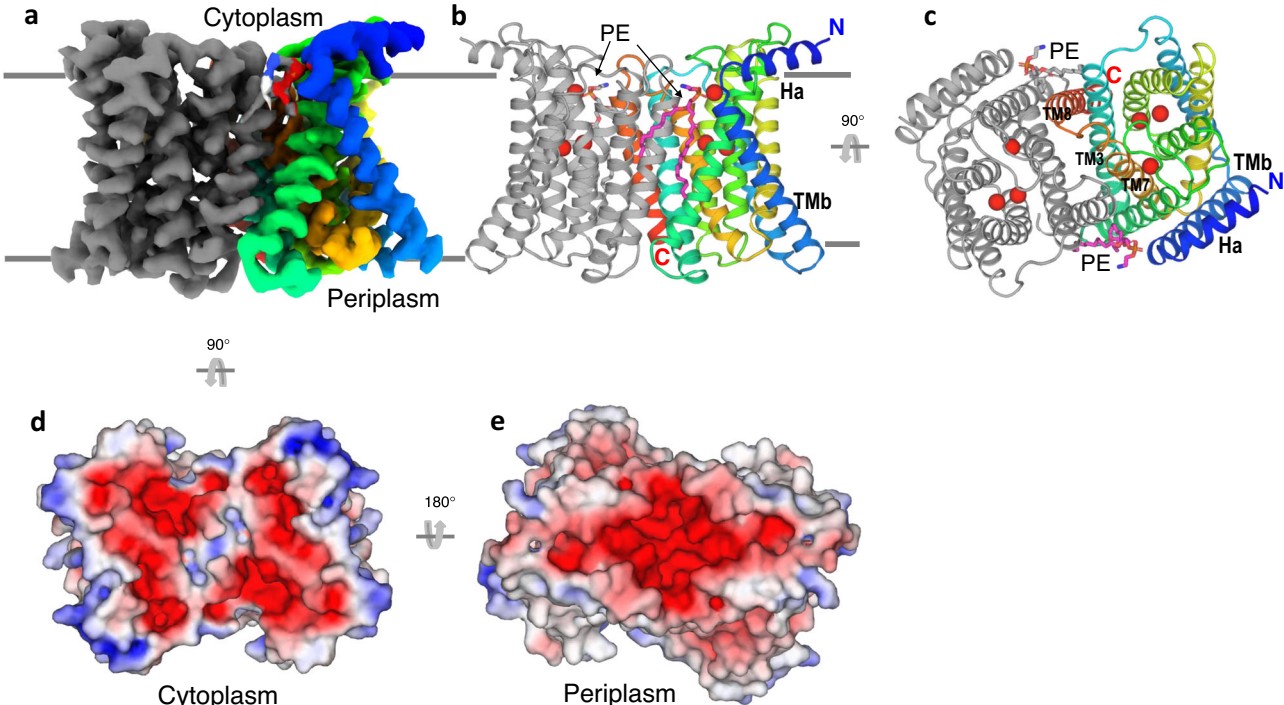

**Fig. 1 | Overall structure of the *Bb*ZIP dimer. a** Cryo-EM density map of the *Bb*ZIP dimer. One subunit was colored in rainbow and the other was colored in gray. **b**, **c** Two views of the overall structure of the dimer. The coloring is as **a**. Six-bound $Cd^{2+}$ ions were shown as red spheres. Two phosphatidylethanolamine (PE) lipids were shown as sticks. **d**, **e** Views of the electrostatic surface of the dimer from the cytoplasm (**d**) and periplasm (**e**). The electrostatics was calculated using the program APBS[60] and plotted at the level of ±5 kT/e. More negative charges are colored redder, and more positive charges are colored bluer.

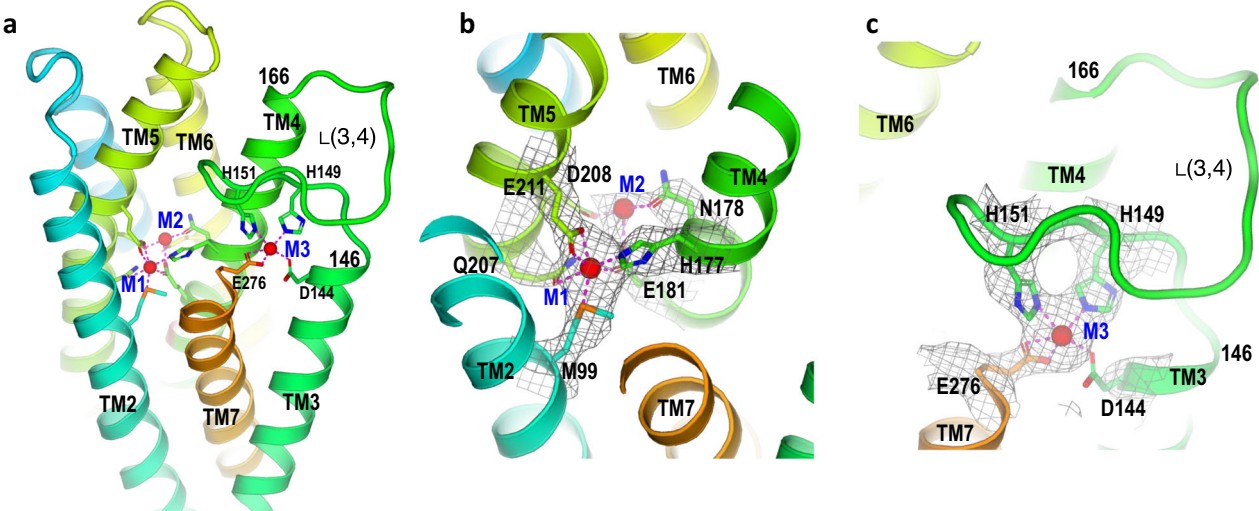

**Fig. 2 | Metal-binding sites in an inward-facing, inhibited state. a** Three metal-binding sites, two $Cd^{2+}$ in the binuclear metal sites M1 and M2 and one close to cytosol (M3). **b** Cryo-EM densities and the interactions for the binuclear metal sites. The $Cd^{2+}$ in the M1 site has much stronger densities (gray isomeshes) than the one in the M2 site. **c** The metal-binding site M3 facing cytoplasm. The M3 site is formed by residues His[149] and His[151] on a loop L (3,4), Asp[144] from TM3, and Glu[276] from TM7.

## Metal binding sites

In each monomer, two $Cd^{2+}$ ions are in the M1 and M2 binuclear sites in the middle of TM2, TM4, and TM5 as previously reported in the crystal structure (Fig. 2a, b)[26]. The third ion binds to a site (M3) near the cytoplasmic surface (Fig. 2a) that consists of residues Asp[144] on TM3, Glu[276] on TM7, and two histidine residues, His[149] and His[151], on a loop (residues 146–166) between TM3 and TM4 (Fig. 2c). We thus named this loop as L(3,4). The loop has alternative conformations: disordered in the crystal structure and ordered in the cryo-EM structure (Supplementary Fig. 6).

The cryo-EM density for the M1-site $Cd^{2+}$ is much higher than the M2-site $Cd^{2+}$, indicating a partial occupancy in the M2 site (Fig. 2b). The M1-$Cd^{2+}$ has well-defined interactions with the side chains of His[177], Asp[211], Gln[207], and Met[99]. In contrast, the M2-site $Cd^{2+}$

The *Bb*ZIP dimer creates three negatively charged surface cavities, two on the cytoplasmic side and one on the periplasmic side (Fig. 1d, e). These negatively charged cavities may have electrostatic interactions with positively charged $Zn^{2+}$ ions, causing the accumulation of ions near the entrance and exit sites.

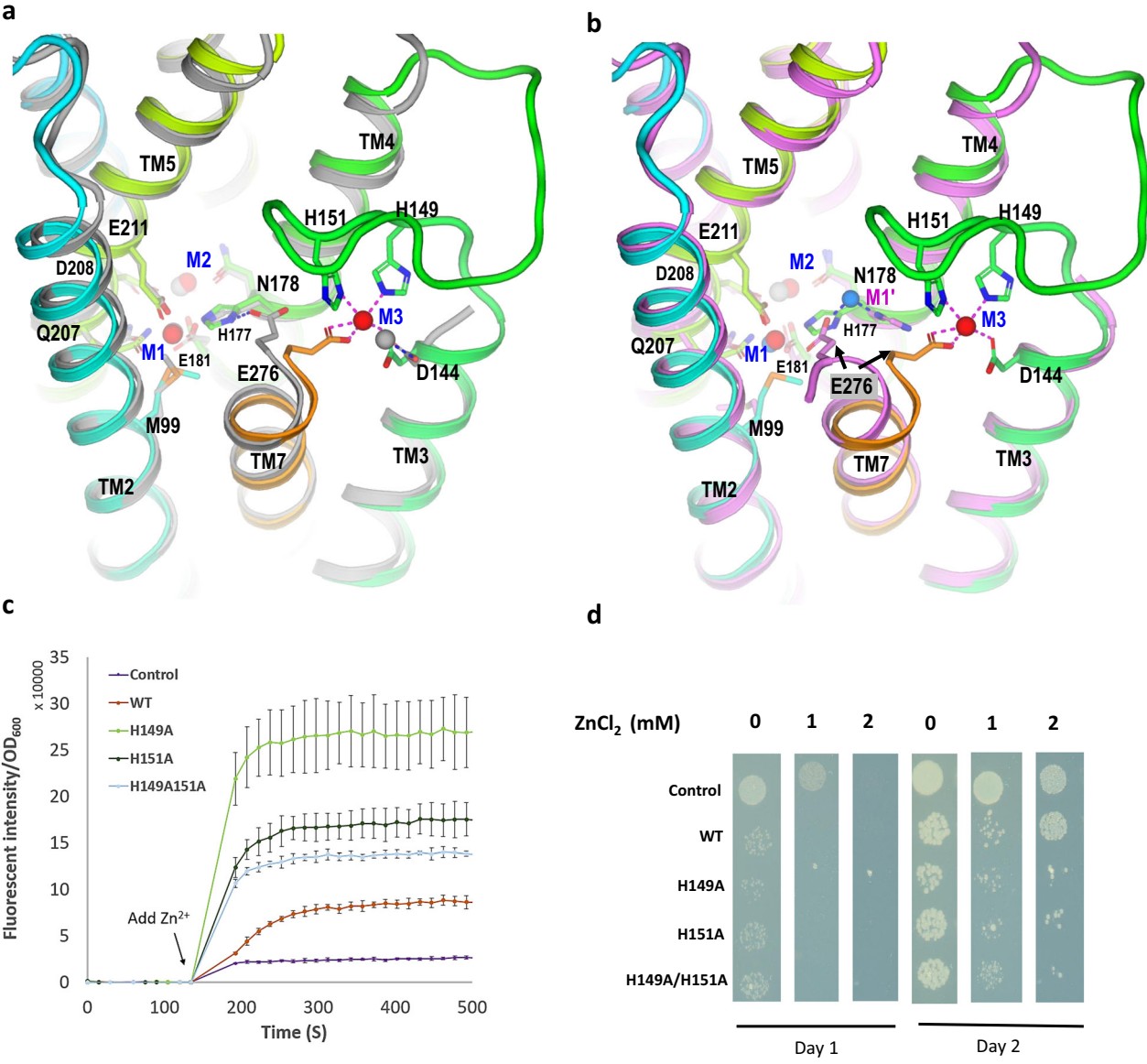

**Fig. 3 | Metal-release and inhibition. a** Superimposition of the metal-binding sites in the cryo-EM structure with the crystal structure (gray color, PDB code 5TSB) bound with $Cd^{2+}$ (gray spheres). **b** Superimposition of the metal-binding sites in the cryo-EM structure with its crystal structure (magenta color, PDB code 5TSA) bound with $Zn^{2+}$ and $Cd^{2+}$. Gray spheres for $Cd^{2+}$ and blue sphere for $Zn^{2+}$. **c** $Zn^{2+}$ uptake inhibited by $His^{149}$ and $His^{151}$ on the loop L(3,4). With the addition of 0.5 mM $Zn^{2+}$

outside, intracellular fluorescence intensity normalized by $OD_{600}$ was measured. Data are presented as mean values ± SD. The error bar is the standard deviation (s.d.) from three independent samples. The experiments were replicated at least three times. **d** Cell growth viability assay with the presence of indicated $Zn^{2+}$. Cells expressing the three mutants on the loop have decreased cell viability. Control, empty expression vector; WT, wild-type.

has poorly defined interactions with $Asp^{208}$, $Asn^{178,}$ and $Glu^{181}$. A recent $Zn^{2+}$ soaking experiment using preformed *Bb*ZIP/$Cd^{2+}$ crystals suggests that the M1-site $Cd^{2+}$ can be replaced by $Zn^{2+}$, but not the M2-site $Cd^{2+}$[31,], thus supporting a notion that the M1-site is more relevant to $Zn^{2+}$ transport. Site-directed mutagenesis for the M2 site indicates that the M2 site is not essential for $Zn^{2+}$ transport[31]. So, the observation of a partial occupancy of $Cd^{2+}$ in the M2 site in the cryo-EM structure is consistent with a less essential role of the M2 site in the $Zn^{2+}$ transport activity. Additionally, the conservation of key residues for the M1-site is higher than for the M2-site, suggesting the M1-site plays an essential role in $Zn^{2+}$ transport. (Supplementary Fig. 7)[6].

### Metal transport and release
To understand the mechanism of metal transport mediated by *Bb*ZIP, we compared the cryo-EM structure with its crystal structure (PDB

code 5TSB) which has three bound $Cd^{2+}$ ions in a crystallized monomer[26]. The alignment of the two structures for the metal-binding sites shows that the binuclear sites are almost the same (Fig. 3a). However, the M3 site at the cytoplasmic side is different. In the crystal structure, $Cd^{2+}$ binds only weakly to $Asp^{144}$, while in the cryo-EM structure, it binds to $Asp^{144}$, $Glu^{276}$, $His^{149}$, and $His^{151}$. We also note that $Glu^{276}$ adopts two different conformations in the crystal vs cryo-EM structure. Its side chain interacts with $His^{177}$ in the crystal structure while it forms part of the M3 site in the cryo-EM structure. Notably, in the crystal structure in which $Cd^{2+}$ was replaced by $Zn^{2+}$ through soaking (PDB code 5TSA), $Glu^{276}$ and $His^{177}$ interact with a $Zn^{2+}$ denoted as M1' site (Fig. 3b). Comparing the different conformations of $Glu^{276}$ in the three structures, it thus appears that $Glu^{276}$ may have a role in the release of $Zn^{2+}$ from the M1-site to the cytosol via the M1' site. $Glu^{276}$ is conserved in the ZIP family at the end of TM7 (Supplementary Fig. 7). To accommodate the release of $Zn^{2+}/Cd^{2+}$ from the M1' site, TM7 also

rotates 1/3 turn so that the $Glu^{276}$ carboxylate group can reach the cytoplasmic side.

## Intracellular $Zn^{2+}$ sensing and regulation

Zinc is toxic and its intracellular concentration must be tightly regulated for proper physiological functions. $His^{149}$ and $His^{151}$ are on the loop L(3,4) which is disordered in the crystal structure but is well ordered in the cryo-EM structure through the formation of the M3 site (Figs. 2c and 3a). We thus hypothesized that the two histidine residues might be part of an intracellular $Zn^{2+}$ sensor that modulates $Bb$ZIP transport activity and prevents $Zn^{2+}$ overaccumulation and cytotoxicity. This hypothesis is consistent with both histidines interacting with $Glu^{276}$ which is critical for $Zn^{2+}$ transport activity. To test this intracellular $Zn^{2+}$ sensing hypothesis, we mutated each of the two histidine residues individually or in combination to alanine residues and measured the $Zn^{2+}$ uptake of these mutants in bacteria overexpressing $Bb$ZIP. To achieve this, we overexpressed wild-type (WT) $Bb$ZIP and its mutants in bacterial cells, loaded the cells with the $Zn^{2+}$ dye FluoZin3, added 500 µM $Zn^{2+}$ outside, and measured the fluorescence intensities (Fig. 3c). Compared with the control cells carrying an empty plasmid or WT, the three histidine mutants showed increased $Zn^{2+}$ uptake as indicated by increased fluorescence intensities upon addition of $Zn^{2+}$. It is thus possible that the mutation of one or both histidine residues impaired the formation of the $Zn^{2+}$-bound M3 site. Consequently, the loop L(3,4) adopts a disordered conformation, promoting $Zn^{2+}$ uptake and release mediated by the $Glu^{276}$ from the M1 site into the cytosol via the M1' site (Fig. 3a, b).

Because the increased $Zn^{2+}$ uptake is toxic to cells, we next tested the toxicity in terms of cell growth viability. We induced the expression of empty plasmid (control), WT, and three histidine mutants in bacteria and plated them on Luria Broth (LB)-Agar plates under different $Zn^{2+}$ conditions. Growth was compared for cells carrying the control empty plasmid, or those expressing either WT or mutants in the absence or the presence of 1 or 2 mM of $Zn^{2+}$ to evaluate cytotoxicity (Fig. 3d and Supplementary Fig. 8). After two days, cells expressing WT $Bb$ZIP can tolerate exposure to 2 mM $Zn^{2+}$ in the medium whereas the three histidine mutants show minimal growth demonstrating that increased $Zn^{2+}$ uptake is toxic. These data provide support for our hypothesis that $Bb$ZIP uses the two histidine residues and loop L(3,4) to sense intracellular $Zn^{2+}$ status and negatively regulate $Bb$ZIP's $Zn^{2+}$ uptake activity when $Zn^{2+}$ is present in abundance.

In addition to $His^{149}$ and $His^{151}$, $Asp^{144}$ on the C-terminus of TM3 also contributes to the formation of the M3 site (Figs. 2c and 3b). In the presence of accumulated intracellular $Zn^{2+}$, the three residues may form a functional $Zn^{2+}$ sensor that locks the side-chain conformation of $Glu^{276}$ in the M3 conformation, thus blocking the $Zn^{2+}$ uptake and release. To form the $Zn^{2+}$ sensor, L(3,4) needs to be twisted through residue $Pro^{148}$ which orients its next residue $His^{149}$ to form the M3 site. We, therefore, propose a $Zn^{2+}$ sensor of a pattern 144DXXXPHXH151, where X can be any residue. Using this pattern, we performed a PHI-BLAST[37] search against the NIH non-redundant sequences and identified 8,436 sequences containing the pattern. Supplementary Fig. 9 shows a reconstructed phylogenetic tree using MEGA11[38] at a level of 90% sequence identity. The tree includes a wide range of the most dangerous human pathogens as well as microorganisms that are capable of cellulose degradation, nitrogen fixation, hydrogen production and oxidation, and methane production. ZIPs in prokaryotes may use a built-in intracellular $Zn^{2+}$ sensor to autoregulate their transport activity to survive in both zinc-deficient and high-zinc toxic environments.

## Mechanism of negative regulation of $Zn^{2+}$ uptake

We propose a model to illustrate how $Bb$ZIP function may be regulated by the intracellular $Zn^{2+}$ sensor in Fig. 4. Although there is a lack of a needed $Bb$ZIP structure in an outward-facing conformation, the negatively charged surface on the periplasmic side of the dimer would attract and accumulate $Zn^{2+}$ through electrostatic interactions (Fig. 1e). Such interactions may help create an outward-facing conformation in which the loop L(3,4) has to be disordered so that $Zn^{2+}$ can reach the M1 site and bind to $Glu^{181}$ and $His^{177}$ (Fig. 4a). Under this scenario when intracellular $Zn^{2+}$ concentration is low, $Glu^{276}$ changes its side chain conformation to receive a $Zn^{2+}$ from the M1 site assisted by $His^{177}$ which also changes its side chain conformation to deliver $Zn^{2+}$ from the M1 site to M1'. Subsequently, $Zn^{2+}$ may be passed from $Glu^{276}$ to $Asp^{144}$ and released into the cytosol (Figs. 3a and 4b). Under intracellular zinc-deficient conditions, the transport cycle will repeat to accumulate intracellular $Zn^{2+}$. When the intracellular $Zn^{2+}$ reaches a threshold which is sensed by $His^{149}$ and $His^{151}$ on L(3,4), the two residues recruit intracellular $Zn^{2+}$ to form the inhibitory M3 site with the two $Zn^{2+}$-releasing residues $Asp^{144}$ and $Glu^{276}$, thus blocking the $Zn^{2+}$ release pathway (Fig. 4c). In addition, the folded loop L(3,4) prevents the conformational switch from inward-facing to outward-facing conformation. Consequently, the transport activity is blocked until intracellular $Zn^{2+}$ concentration is below the threshold through mobilization and utilization of intracellular $Zn^{2+}$ by $Zn^{2+}$ binding proteins or $Zn^{2+}$ efflux transporters[6,39]. It's noted that additional structures are needed to define the relationship between the L(3,4) loop positioning and $Zn^{2+}$ concentrations.

## Discussion

In this work, we determined a cyro-EM structure of a ZIP transporter dimer that supports its function as a transporter through a proposed two-domain elevator model[27]. However, previously solved crystal structures were determined as a monomer[26]. We note that crystal structures do not contain the N-terminal Ha and TMb, and loops between TM1-2 and TM3-4 (L(1,2) and L(3,4), respectively) are disordered (Supplementary Fig. 6). Loop L(1,2) is located on the interface of the $Bb$ZIP dimer and its disordering in crystal structures might affect the stability of the dimer. In addition, two lipids identified in the cryo-EM structure may contribute to the dimerization, which was disrupted in crystal structures (Fig. 1b, c). $Bb$ZIP function in the cell may require a dimer to stabilize the two-domain elevator model (Fig. 4). In humans, ZIP4 and ZIP13 have been shown as homodimers[40,41], and ZIP6 and ZIP10 are suggested to form heterodimers to regulate embryonic development and cell migration[42]. Although the functional significance of $Bb$ZIP and other ZIP dimers remains to be determined, one possible function of the dimerization is to make ZIPs more stable during conformational transitions that are required for $Zn^{2+}$ uptake. Computational simulation suggests that monomeric $Bb$ZIP is unstable in an outward-facing conformation[43]. Therefore, the $Bb$ZIP dimer structure provides a basis to understand the transport mechanism of the ZIP family.

In the cytoplasm, physiological $Zn^{2+}$ is maintained at a very low concentration i.e., in the nanomolar range ($10^{-9}$ M). In adaptation to environmental $Zn^{2+}$ deficiency or toxicity, some living organisms may have evolved a mechanism to sense their intracellular $Zn^{2+}$ status. We propose that $Bb$ZIP has a built-in $Zn^{2+}$ sensor (DXXXPHXH) that negatively regulates its own $Zn^{2+}$ uptake activity. The sensor integrates two histidines ($His^{149}$ and $His^{151}$) on L(3,4) and two carboxylates on $Asp^{144}$ and $Glu^{176}$. Because $Asp^{144}$ and $Glu^{176}$ are also part of the $Zn^{2+}$ transport pathway, conformational changes of the loop L(3,4) enable $Bb$ZIP to regulate its activity through the detection of the intracellular $Zn^{2+}$ status. In our $Zn^{2+}$ uptake experiments, mutating either histidine residue on L(3,4) resulted in the accumulation of more $Zn^{2+}$ inside, consistent with a compromised negative regulation (Fig. 3c). For comparison, wild-type $Bb$ZIP allows uptake of less $Zn^{2+}$ through a stronger binding of $Zn^{2+}$ to the M3 site. For both wild-type and mutants, we found that intracellular $Zn^{2+}$ reached saturation in about 200–300 s after zinc addition. It is thus possible that the accumulated zinc may not only bind to L(3,4) and the M3 site, but also residues in the M1 and M1' sites to suppress the uptake of additional $Zn^{2+}$. Moreover, $Zn^{2+}$ binding to these sites may contribute to the lock of a $Bb$ZIP

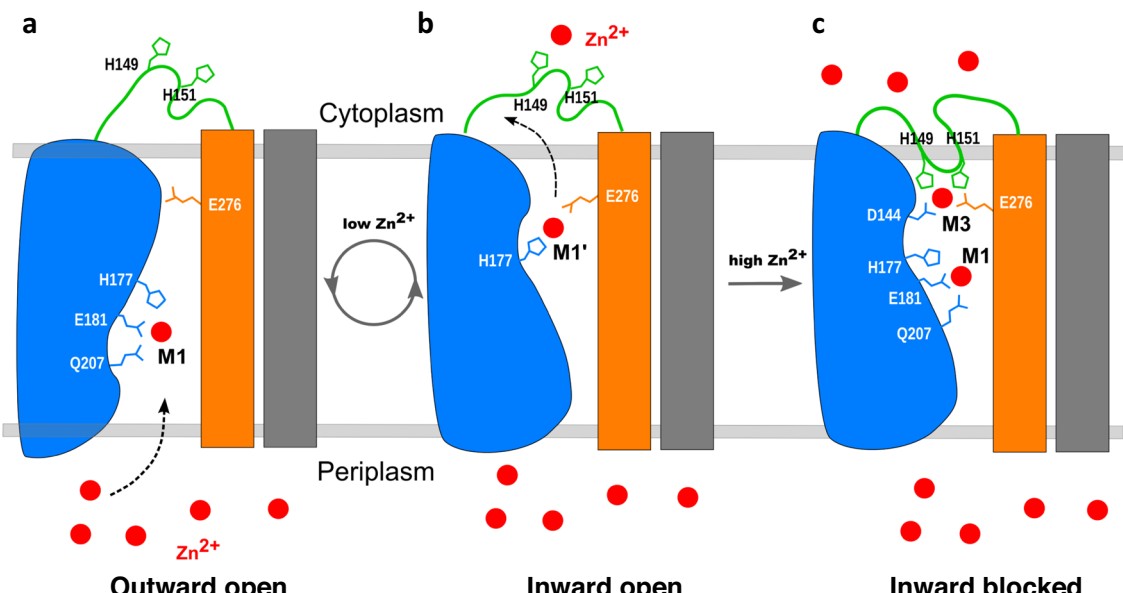

**Fig. 4 | Proposed mechanism of intracellular Zn²⁺ sensing and regulation. a** In the presence of extracellular Zn²⁺, *Bb*ZIP will have an assumed outward-facing conformation in which Zn²⁺ has access to a binding site (M1) involving residues Glu¹⁷⁷, His¹⁸¹, and Gln²⁰⁷. The access of Zn²⁺ to the M1-site triggers a conformational switch from outward facing to an inward-facing conformation with the Zn²⁺ remaining in the M1-site as shown in Fig. 3a. **b** The side chain of His¹⁷⁷ changes its conformation and delivers a Zn²⁺ ion to Glu²⁷⁶ and Asp¹⁴⁴. Zn²⁺ is then released to cytosol when intracellular Zn²⁺ is low. **c** With the accumulation of intracellular Zn²⁺, His¹⁴⁹ and His¹⁵¹ function as a Zn²⁺ sensor together with Asp¹⁴⁴ to form the M3-site which inhibits the further Zn²⁺ release and conformation flipping from inward-facing to outward-facing for another transport cycle. Blue is the elevator motion domain consisting of TM2 and TM4-6; orange is a scaffold consisting of TM3 and TM7-8. Gray is the scaffold domain from another monomer. The elevator motion domain from the other monomer is omitted for clarity.

structure in the inward-facing conformation which blocks the access of external Zn²⁺ (Fig. 4). This zinc-sensor-lock mechanism is consistent with conformational changes previously reported for a Zn²⁺-free *Bb*ZIP structure²⁷. The motif we identify for this sensor is highly conserved in bacteria and archaea, suggesting this mode of negative regulation represents an ancient homeostatic mechanism that evolved before the divergence of prokaryotes and eukaryotes (Supplementary Fig. 9). In eukaryotes, many ZIPs harbor a histidine-rich loop between TM3 and TM4. For example, *Arabidopsis* IRT1 uses the loop to sense and regulate its endocytosis and degradation³². Human ZIP4 uses a histidine-rich cluster to mediate ubiquitination and degradation upon Zn²⁺ accumulation³⁰,⁴⁴. Although there is a lack of structure for a eukaryotic ZIP, a similar foldback and negative regulation mechanism might exist in some plant and human ZIPs.

It was proposed that *Bb*ZIP may function as a pH-sensitive and water-regulated channel²⁸,⁴⁵. However, our dimer structure of *Bb*ZIP is more consistent with a two-domain elevator transporter model²⁷. In our cryo-EM structure, TM3, TM7, and TM8 form a stabilized dimer interface, and TM2 and TM4-6 form the elevator domain that changes conformation while switching between outward-facing and inward-facing to facilitate Zn²⁺ binding and release through an alternative access model (Fig. 4).

The uptake of Zn²⁺ by ZIPs is pH dependent. For example, human ZIP2 is pH-dependent with increased activity at a higher pH⁴⁶. Human ZIP4 was proposed to be a Zn²⁺/H⁺ symporter⁴⁷. For *Bb*ZIP, a pH-sensitive Zn²⁺ uptake mechanism was proposed²⁸. We were curious whether the transport activity of the *Bb*ZIP dimer is also pH dependent. These reports prompted us to load the pH-sensitive dye BCECF (2′,7′-bis-(carboxyethyl)-5-(and-6)-carboxyfluorescein) to cells expressing WT or histidine mutants. We added Zn²⁺ outside to initiate uptake, and measured intracellular pH using BCECF fluorimetry. As shown in Supplementary Fig. 10, upon the addition of Zn²⁺, intracellular pH increased with the largest increase from histidine mutants compared with WT or the empty vector control. The pH increase correlates roughly with the accumulation of intracellular Zn²⁺ (Fig. 3c and

Supplementary Fig. 10). When Zn²⁺ uptake reached saturation, intracellular pH also stopped increasing, suggesting a linkage between Zn²⁺ uptake and H⁺ efflux. We thus propose that *Bb*ZIP may function as a Zn²⁺/H⁺ antiporter, reassembling ZnTs which export Zn²⁺ in exchange for H⁺ influx⁴⁸. Bacterial intracellular pH is maintained at about pH 7.5. Our pH measurements at an external pH of 7.5 did not cause a proton gradient or proton motive force. Therefore, the increased intracellular pH was likely induced by Zn²⁺ influx in exchange for H⁺ efflux.

## Methods
### Protein expression and purification
The gene encoding the full length of the zinc transporter from *Bordetella bronchiseptica* (*Bb*ZIP) was synthesized and cloned into pET15b expression vector by Genscript (Piscataway, NJ, USA). The vector contains an N-terminal 6×Histag followed by a thrombin cleavage site. Protein expression was performed using *Escherichia coli* C43 (DE3) cells (Lucigen, Middleton, USA) growing in an auto-induction culture medium at 37 °C. Cells were harvested by centrifuge at 5000 × *g* for 10 min at 4 °C and pellets were resuspended in lysis buffer consisting of 30 mM HEPES, pH 7.5, 300 mM NaCl, 2 mM MgCl₂, and 0.5 mg lysozyme per gram of cell. Cells were lysed using an EmulsiFlex-C3 homogenizer (Avestin, Ottawa, Canada) at 15,000 psi. Cell lysates were cleaned by centrifugation at 17,000 × *g* for 25 min at 4 °C and the supernatants were collected for pelleting membranes by ultracentrifugation at 250,000 × *g* overnight at 4 °C.

Membranes were resuspended in solubilization buffer (20 mM HEPES, pH 7.3, 300 mM NaCl, 0.5 mM CdCl₂, 4% DDM (w/v), and Roche protease inhibitor cocktail). After solubilization for 4 h at 4 °C, the insoluble components were removed by ultracentrifugation at 310,000 × *g* for 30 min at 4 °C, and the supernatant was diluted with a buffer (20 mM HEPES, pH 7.3, 300 mM NaCl) supplemented with 20 mM imidazole, and applied to a pre-equilibrated Ni²⁺-NTA column (Anatrace, Maumee, OH, Cat# SUPER-NINTA100). The column was washed by 10 column volumes (cv) of wash buffer (20 mM HEPES, pH 7.3, 300 mM NaCl, 6.25% glycerol (w/v), 50 mM imidazole, 0.1% DDM,

50 μM CdCl$_2$). The protein was eluted with 300 mM imidazole, 0.05% DDM, and the 6xHistag was cleaved by thrombin overnight at 4 °C. The imidazole concentration in protein was lowered to less than 20 mM by concentrate-dilution with a 50 kDa molecular cutoff concentrator (Millipore Sigma, Burlington, MA, USA). Proteins were passed through the pre-equilibrated Ni$^{2+}$-NTA column and the flow-through was concentrated to 10 mg/mL and further purified by size exclusion chromatography using a Superdex 200 Increase column (GE Healthcare, Inc, Chicago, IL, USA) in a buffer containing 10 mM HEPES, pH 7.25, 300 mM NaCl, 6.25% glycerol (W/V), and 0.02% DDM. CdCl$_2$ was added to each collected fraction to a final concentration of 0.25 mM. Fractions containing the protein were pulled together and concentrated to 21 mg/mL using a 50 kDa molecular cutoff concentrator.

## Amphipol reconstitution

The purified protein (21 mg/mL) in a buffer (10 mM HEPES, pH 7.25, 300 mM NaCl, 6.25% glycerol, 0.02% DDM (w/v), and 0.25 mM CdCl$_2$) was mixed with amphipol PMAL-C8 (Anatrace, Maumee, OH, Cat# P5008) in a mass ratio of protein/amphipol of 1:5 (w/w). The mixture was incubated at 4 °C for 20 h and concentrated using a 50 kDa molecular weight cutoff concentrator. Then, SM-2 Bio-beads (60 mg wet, Bio-Rad, Cat# 1523920) were added to the mixture, and the mixture was nutated at 4 °C for 3 h. Prior to use, Bio-Beads were soaked with buffer containing 20 mM HEPES, pH 7.6, and 100 mM NaCl overnight at 4 °C. The Bio-beads were removed by passing through a micro Bio-Spin chromatography column. The flow-through containing the reconstitution mixture was centrifugated at 20,000 × g for 1 h at 4 °C; and the supernatant was used for size-exclusion chromatography through a Superdex 200 Increase column (GE Healthcare, Inc., Chicago, IL, USA) in a buffer containing 20 mM HEPES, pH 7.5 and 100 mM NaCl. The fractions containing the *Bb*ZIP-PMAL-C8 nanodiscs were collected and concentrated to 4 mg/mL for electron microscopy analyses.

## Zinc uptake assay in bacteria

All mutant constructs using pET15b were produced by site-directed mutagenesis following the manufacturer's protocol and verified by DNA sequencing. The expression constructs were transformed into C43 (DE3) competent cells. Single colonies were inoculated into 6 mL Terrific Broth (TB) media at 37 °C for 3 h. When the bacterial cell culture reached an OD$_{600}$ value of between 0.4–0.6, the temperature was lowered to 25 °C and the expression was induced by adding IPTG at a final concentration of 0.33 mM for 20 h. The cells transformed with the wild-type or empty pET15b plasmids were used as positive and negative controls, respectively. Cells were pelleted by centrifugation at room temperature (3000 × g, 5 mins). The cell pellets were then washed with wash buffer (20 mM Tris-HCl, pH 7.5, 150 mM KCl). The cell pellets were resuspended in 500 μL wash buffer supplemented with zinc fluorescence dye FluZin3-AM (Thermo Fisher Scientific, Waltham, MA, Cat# F24195) to a final concentration of 5 μM. Cells were incubated at 37 °C in dark for at least 1 h. Cells were then washed three times in the wash buffer to remove free dyes by repeated suspension in the wash buffer and centrifugation. Cell pellets were finally suspended in 1 mL wash buffer for Zn$^{2+}$ uptake assay.

Cell suspensions were aliquoted as 200 μL each vial. After centrifugation, the cell pellets were resuspended in an assay buffer of 20 mM Tris-Bis-HCl, pH 7.5, and 150 mM KCl. 90 μL of cell suspensions were added to a well of 96-well Greiner black-bottom plates with three replicates. After reading at least ten data points, 10 μL of 5 mM ZnCl$_2$ was added to each well (final Zn$^{2+}$ concentration 500 μM). Fluorescence intensity signals were measured using a Tecan Spark 20 M plate reader (Tecan US, Inc., Morrisville, NC, USA).

For measuring the fluorescence emission of FluoZin3, the excitation/emission wavelength was set to 515/535 nm with a 7.5 nm filter window. The gain was manually set to 150%. The temperature was set to 37 °C. A total duration of 8 min was divided into 32 cycles with 15-s

intervals. The zinc uptake was monitored as the increased fluorescence after adding zinc normalized by cell density of OD$_{600}$.

Primers used are H149A forward: GACTACTTCACGCCGGCCGA GCACGAACGCACC, reverse: GGTGCGTTCGTGCTCGGCCGGCGTGA AGTAGTC; H151A forward: GAGGCCGAACGCACCGGCCACCAGG, reverse: TTCGGCCTCGTGCGGCGTGAAGTAGT; H149A-H151A forward: GAGGCCGAACGCACCGGCCACCAGG, reverse: TTCGGCCTCGGCCGG CGTGAAGTAGT.

## Intracellular pH measurement

To measure intracellular pH changes, we loaded cells with BCECF-AM (ThermoFisher Scientific, Waltham, MA, Cat# B1170) following the manufacturer's manual. After three washes using the assay buffer, 90 μL cells were loaded to each well for 200-s basal fluorescence measurement. Then 0.5 mM ZnCl$_2$ was added to each of the wells to initiate Zn$^{2+}$ uptake. For the fluorimetry measurement, the excitation wavelengths were set at 440/490 nm with a 10 nm filter window. The emission wavelength was set at 535 nm with a 10 nm filter window. The gain was manually set to 120%. The temperature was set to 37 °C. A total of 62 cycles of measurements were taken at 10-s intervals. After that, 0.5% DDM was added, and another 60 cycles of measurements were taken at 10-s intervals.

## Bacterial growth viability assay

The expression constructs were transformed into C43 (DE3) pLysS competent cells. Three colonies were inoculated into 6 mL of Terrific Broth (TB) media at 37 °C. When the OD$_{600}$ value reached between 0.5–0.6, 0.33 mM IPTG (final) was added to induce the expression at 25 °C for 20 h. Cells were pelleted by centrifugation at 3000 × g for 5 min at room temperature. The pellets were resuspended in LB media adjusted by OD$_{600}$ values. Suspended cells were diluted to OD$_{600}$ of 0.1, 0.01, 0.001, 0.0001, 0.00001, and 0.000001. Four microliters of diluted cells were placed on LB-Agar plates supplemented with 0.2 mM IPTG and various concentrations of ZnCl$_2$ (0, 1, and 2 mM). The plates were placed at room temperature for bacterial growth.

## Cryo-EM sample preparation and data collection

Three microliters of the *Bb*ZIP-PMAL-C8 nanodiscs were applied to a glow-discharged (15 mA current for 15 s) 300-mesh R 0.6/1 UltrAuFoil Holey Gold grid (Electron Microscopy Sciences, Hatfield, PA, Cat# Q350AR1A). After waiting for about 60 s, vitrification was performed using a ThermoFisher Mark IV vitrobot (ThermoFisher Scientific, Waltham, MA) with a blotting condition of 5 s blot time, 0 blot force, and 100% humidity at 6 °C.

Single-particle cryo-EM data were collected at the LBMS facility at Brookhaven National Laboratory using a ThermoFisher Titan Krios electron microscope (G3i) equipped with a Gatan K3 camera and a BioQuantum energy filter. With a physical pixel size of 0.666 Å (0.333 Å in super-resolution), a total dose of 64 e$^-$/Å$^2$ was fractioned to 50 frames using the ThermoFisher data acquisition program EPU. A total of 6909 movies were collected with an energy filter width of 15 eV throughout the data acquisition. Data collection statistics are listed in Supplementary Table 1.

## Cryo-EM data processing

Fractioned movies were corrected using MotionCorr2[49] with a binning factor of 2. Corrected and averaged movie frames were further corrected by CTF estimation using Gctf[50]. Micrographs with an estimated resolution better than 4.5 Å were selected for particle picking. Particles were initially picked and cleaned up by 2D class averaging in Relion3[51]. Then the cleaned particles were used for training and particle picking using Kpicker[52]. A total of 1,997,192 particles were picked, extracted at 256 pixels, and binned to 64 pixels with a pixel size of 2.664 Å.

We used cryoSPARC[53] for particle clean-up through 2D class averaging which resulted in 1,901,053 selected particles. With four

classes each, selected particles were used for two cycles of ab initio reconstruction followed by 3D heterogenous refinements with a particle size of 64 pixels (2.664 Å). The process ended up with 497, 896 particles which were centered and re-extracted at 256 pixels and binned to 128 pixels with a pixel size of 1.332 Å. The re-extracted particles were used for 3D heterogeneous refinement using 3 classes from which the best class containing 177,542 particles was selected, re-centered, and re-extracted at 256 pixels, and binned to 128 pixels (pixel size 1.332 Å) (Supplementary Fig. 3). No symmetry was used during the ab initio model generation and heterogenous refinements in cryoSPARC.

Re-extracted particles were auto-refined to convergence in Relion3[51] followed by non-alignment 3D classification into three classes without symmetry (Supplementary Fig. 3). Particles from the 3D class with the best structural feature (α helices and side chain features) as visualized in Chimera[54] were selected. 43,498 particles were selected for CTF refinement in Relion3 followed by 2D classification and non-uniform refinement in cryoSPARC to reach a reconstruction at 3.05 Å in C2 symmetry based on gold-standard Fourier Shell Correlation (Supplementary Fig. 4d). Local resolutions were estimated using BlocRes[55]. Reconstruction statistics are listed in Supplementary Table 1.

### Model building and refinement
To boost the side-chain features of the map, the masked and filtered map was sharpened using PHENIX[56] with a $B$ factor of −41 Å$^2$. The solved crystal structure of *Bb*ZIP (PDB code 5TSB) was used as a starting model to build the dimeric structure in COOT[57]. All refinements were performed in real space in PHENIX. The refined model was validated using Molprobity[58] and the refinement statistics are listed in Supplementary Table 1.

### Model visualization
The cryo-EM density map and atomic models were visualized using either PyMOL[59] or Chimera[54].

### Reporting summary
Further information on research design is available in the Nature Portfolio Reporting Summary linked to this article.

## Data availability
The data that support this study are available from the corresponding authors upon request. The 3-dimensional cryo-EM density map has been deposited in the Electron Microscopy Data Bank (EMDB) under the accession code EMD-40050 (*Bb*ZIP). Atomic coordinates have been deposited in the Protein Data Bank (PDB) under the accession number 8GHT (*Bb*ZIP). Related PDB codes used in this study can be found at 5TSA and 5TSB. The source data underlying Fig. 3c, Supplementary Fig. 1a, b, Supplementary Fig. 4d, and Supplementary Fig. 10 are provided as a Source Data file. Source data are provided with this paper.

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

## Acknowledgements

We thank Sean McSweeney for critically reading the manuscript and LBMS staff for the help with the cryo-EM operation and data acquisition. This work was supported by the U.S. Department of Energy (DOE), Office of Biological and Environmental Research as part of the Quantitative Plant Science Initiative at Brookhaven National Laboratory. Protein expression purification and sample preparation by J.C. and J.S. were supported by the U.S. Department of Energy, Office of Science, Office of Basic Energy Sciences under contract number DE-SC0012704, specifically through the Physical Biosciences program of the Chemical Sciences, Geosciences, and Biosciences Division. The work used Laboratory for Biomolecular Structure (LBMS) which is supported by the U.S. Department of Energy, Office of Science, Office of Biological and Environmental Research.

## Author contributions

Q.L. designed the study and experiments. P.C. performed biochemistry and cell-based experiments; J.C. prepared samples for structural biology; P.Z. performed protein expression tests and initial zinc uptake experiments; Q.L. performed structural biology work. P.C., J.C., J.S., and Q.L. analyzed the data. Q.L. wrote the manuscript with help from other co-authors.

## Competing interests

The authors declare no competing interests.
