## [Peer Review File · Nature Communications]

Structural mechanism of intracellular autoregulation of zinc uptake in ZIP transportersReviewers' Comments:

Reviewer #1:

Remarks to the Author:

The manuscript is quite interesting and provide new and relevant information on how ZIPs may transport Zn. More importantly, authors have provided some evidence that the histidine-containing cytoplasmic loop could be a Zn sensor that blocks Zn uptake and avoid excessive Zn concentrations in the cytoplasm. This information is now and quite relevant to the field.

I have some concerns and comments over the data presented and the text itself, which are listed below:

- Page 2, line 1: please correct "actions".
- Page 2, line 12-15: this section, although correct, should be more comprehensive regarding the up-to-date literature in plants. ZIP transporters have been considered the likely Zn²⁺ uptake proteins from the soil by root cells. But primary evidence only appear in 2020, in a series of papers describing OsZIP9. Please consider citing the relevant papers (10.1104/pp.19.01569, 10.1104/pp.20.00125, 10.1111/tpj.14855).
- Page 2, line 15: here I think authors say ZIPs are regulated translationally. I am not sure what they want to say, but ZIPs are usually regulated by Zn deficiency at the transcriptional level. I do not know of any description of post-transcriptional regulation for ZIPs (except for IRT1, but by Fe – not the same context). Please correct or re-write to be more precise.
- Page 2, line 23-29: this paragraph should be slightly clearer that this is previous work – it provided some insight into ZIPs structure, but some details were not clear and the current work improves that. This information is given at the beginning of the results section. I suggest reorganizing to improve clarity.
- Page 3, line 5-6: "and little is known about the regulation of ZIP function.". Authors mean that little is known about the intracellular loop function in regulating ZIP function, correct? Please rephrase for clarity.
- Supp Fig 1: please provide the details of the SDS-PAGE – what kind of dilutions were used in each lane?
- Supp Fig 2: I am not a specialist in Cryo-EM, but wanted to point out that I could not see the structures mentioned in the text in the Figure. Might be due to the resolution of the image in the reviewing material, but I'll leave that authors to reply.
- Page 5, line 1-2: could authors explain why Cd was used? Why not use Zn instead? This could also be stated in the results section.
- Page 5, line 21-22: What does that mean functionally? Can the site have its Cd replaced by Zn, although that does not happen in the crystal? Please make it clearer to the average reader.
- Page 5, line 22-23: the phrase needs rewriting (please check it), although the meaning is understandable. However, I would suggest that authors to provide alignments to support the claim, instead of only citing a reference.
- Page 6, Line 4: please provide an alignment to support the claim of Glu276 conservation.
- Page 6, line 13: "consistent"- please correct.
- General comments on the model: the idea that the cytoplasmic, histidine-containing loop, works as a inhibitory module to avoid Zn toxicity is interesting, and the experimental data is consistent with that. However, it feels like other explanations are possible, since changes in movement and/or positioning of the loop under different Zn concentrations is not shown in the current paper. Molecular dynamics could help with that, and may be necessary to actually demonstrate the validity of the model.
- Page 8, line 12-13:
- Page 8, line 14-16: this statement agrees with the idea that molecular dynamics might be needed to investigate the model and provide more evidence for its validity.
- Page 8, line 18-19: this claim is still a possibility. I would de-emphasize it.

Reviewer #2:

Remarks to the Author:

Zinc is an essential micronutrient that supports all living organisms through regulating numerous biological processes. However, the function and mechanism of uptake regulation by intracellular Zn²⁺ status are not fully understood. In this manuscript, Pang et al. reported the first cryo-electron microscopy structure of bacterial ZIP transporter from *Bordetella bronchiseptica* (BbZIP) at 3.06 Å. The obtained cryo-EM structure revealed that BbZIP adopted a dimeric configuration with each protomer in an inward facing, inhibited conformation. More importantly, a metal ion egress site was observed to be covered by the loop L(3,4) in the reported cryo-EM structure. A novel mechanism of autoregulating zinc uptake through a built-in sensor has also been proposed and validated biochemically.

Points:

1. P3, L31-32 "To enhance the signal to noise ratio of BbZIP images, we used a physical pixel size of 0.666 Å/pixel in super-resolution mode".

The final reconstructed map was resolved at 3.06Å resolution which is far from the Nyquist (1.32 Å). Authors should explain why collecting images at a much higher magnification perhaps be beneficial to small membrane protein. Otherwise, it would be misleading to the general cryo-EM practitioners.

2. P4, L8-12 "Due to the presence of amphipols around BbZIP, standard ab initio 3D reconstruction in CryoSPARC 3.2 did not produce reasonable structural models. To make use of the structural features of transmembrane helices in ab initio reconstruction, we used data with resolution cutoffs of between 7 and 10 Å to generate four ab initio models from 50,000 randomly selected particles in CryoSPARC." Authors should provide a full gallery of 2D class averages and label the classes selected for ab initio models. Please explain the rationale in more detail why resolution cutoffs (7-10 Å) are applied. Is there any possibility that additional conformational states might be missing due to such a restricted resolution cutoff ?

Authors performed data processing interchanging between Relion and cryoSPARC. However, Suppl Fig.3 is over simplified and extremely confusing. Authors should provide a comprehensive data processing workflow and update it in Suppl Fig.3.

3. P4, L23-27 "In the final reconstruction, most particles represent a wide-spread distribution in the Euler space with slightly side-view particles (Supplementary Fig. 4e). The reconstructed map uses a 2-fold symmetry (C2) which displays a higher resolution than a separate reconstruction with no symmetry (C1) at a resolution of 3.3 Å (Supplementary Fig. 4d)."

It appears that the orientation plot shown in Suppl Fig.4e was from Relion. However, the final reconstruction FSC curve is obtained from cryoSPARC according to the corresponding text in page 21, which is confusing. Authors should report which FSC curves are actually shown in Suppl Fig.4d, masked or unmasked? In addition, FSC curve should be plotted all the way to Nyquist frequency (1.32 Å) not stop at 3Å.

4. P5, L1-2 "We used cadmium in our protein purification; we found six Cd²⁺ ions in the dimer." Authors should carefully examine the C1 reconstruction map to validate the number of Cd²⁺ ions within a dimer. It would be informative to validate indeed the two protomers are in identical conformation in C1 map. Therefore, symmetry expansion followed by 3D classification without alignment should be performed to identify potential intermediate states.

5. P5, L12-16 "The third ion binds to a site (M3) near the cytoplasmic surface (Fig. 2a) that is consisted by residues Asp144 on TM3, Glu276 TM7, and two histidine residues, His149 and His151, on a loop (residues 146-166) between TM3 and TM4 (Fig. 2c). We thus named this loop as L(3,4). The loop has alternative conformations: disordered in the crystal structure and ordered in the cryo-EM structure (Supplementary Fig. 6)."

Solving the loop L(3,4) structure is the main structural discovery in this work. Loop L(3,4) should be

highlighted in the local resolution map indicating the feasibility for building an atomic model particularly at His149 and His151.

6. P7, L14-31 "Mechanism of negative regulation of Zn²⁺ uptake".

The focus and main findings in the current manuscript solely support a potential mechanism that Zn²⁺ uptake could be negatively regulated at the inhibitory M3 site.

It lacks sufficient evidence with vast gap knowledge to explain the mechanism of BbZIP Zn²⁺ uptake through multiple hypothetical intermediate states during an entire cycle. It has been reported that elevator domain undergoes significant conformational changes as a rigid-body relative to scaffold domain upon substrate binding. The illustrated inward open state in Fig.4b is perhaps over simplified. Authors should revise Fig.4b in the updated manuscript.

7. P13, L27 "Data availability"

Author should deposit C1, C2 maps and the sharpened map for modeling in the EMDB. Authors should deposit meta data to EMPIRE upon publication.

8. P21, L13-16 "A total of 1,197,192 particles were picked, extracted at 256 pixels, and binned to 64 pixels with a pixel size of 2.664 Å. We used CryoSPARC 51 for particle clean-up through 2D class averaging which resulted in 1,901,053 selected particles."

Picked number of particles is less than the number of particles after cleaning up. It doesn't make sense.

In summary, Pang et al. solved the first cryo-EM structure of BbZIP transporter in dimeric conformation. The structural findings together with the biochemistry data support a novel mechanism of negative regulation of Zn²⁺ uptake through a built-in sensor. The obtained cryo-EM map looks impressive, however the structural findings behind are less exciting. The inward facing inhibited state structure is likely more homogenous and resolved from only a subset of particles. It is worth to investigate the full dataset more carefully, perhaps identify intermediate states with each protomer as the mechanistic interpretation could be drastically changed.

Reviewers' comments:

Reviewer #1 (Remarks to the Author):

The manuscript is quite interesting and provide new and relevant information on how ZIPs may transport Zn. More importantly, authors have provided some evidence that the histidine-containing cytoplasmic loop could be a Zn sensor that blocks Zn uptake and avoid excessive Zn concentrations in the cytoplasm. This information is now and quite relevant to the field.

Response: We thank the reviewer for the positive evaluation of the significance and contribution of our work.

I have some concerns and comments over the data presented and the text itself, which are listed below:

- Page 2, line 1: please correct “actions”.

Response: Corrected.

- Page 2, line 12-15: this section, although correct, should be more comprehensive regarding the up-to-date literature in plants. ZIP transporters have been considered the likely Zn²⁺ uptake proteins from the soil by root cells. But primary evidence only appear in 2020, in a series of papers describing OsZIP9. Please consider citing the relevant papers (10.1104/pp.19.01569, 10.1104/pp.20.00125, 10.1111/tbj.14855).

Response: Thanks for bringing our attention to recent reports of zn²⁺ uptake by root cells. We now cite these papers in the revised manuscript. (p 2, lines 16-17)

- Page 2, line 15: here I think authors say ZIPs are regulated translationally. I am not sure what they want to say, but ZIPs are usually regulated by Zn deficiency at the transcriptional level. I do not know of any description of post-transcriptional regulation for ZIPs (except for IRT1, but by Fe – not the same context). Please correct or re-write to be more precise.

Response: Thank you for pointing out this typo. We agree with the review that ZIPs are usually regulated by Zn deficiency at the transcriptional level. We have revised the sentence to use “transcriptionally”. (p 2, line 17)

- Page 2, line 23-29: this paragraph should be slightly clearer that this is previous work – it provided some insight into ZIPs structure, but some details were not clear and the current work improves that. This information is given at the beginning of the results section. I suggest reorganizing to improve clarity.

Response: We have added some details about the previous structural work and lack of understanding with respect to its regulatory mechanism. We reorganized this point, moving it from the results section to the introduction. (p. 2, lines 31-33; p. 3, line 1)

- Page 3, line 5-6: “and little is known about the regulation of ZIP function.”. Authors mean that little is known about the intracellular loop function in regulating ZIP function, correct? Please rephrase for clarity.

Response: The reviewer is correct. We have rewritten the sentence to clarify that “little is known about the intracellular loop in regulating ZIPs. (p. 3, lines 11-12)

- Supp Fig 1: please provide the details of the SDS-PAGE – what kind of dilutions were used in each lane?

Response: We have added details of the SDS-PAGE in the Supp Fig. 1 caption. The lanes represent eluted fractions from the initial size-exclusion chromatography.

- Supp Fig 2: I am not a specialist in Cryo-EM, but wanted to point out that I could not see the structures mentioned in the text in the Figure. Might be due to the resolution of the image in the reviewing material, but I'll leave that authors to reply.

Response: We agree with the reviewer that the resolution for the image in Supp Fig 2 is too low. For the cryo-EM method to work, thousands of such particles have to be aligned and averaged to obtain a high-resolution 3D map as shown in Supp Fig. 3 and Fig. 1. We have added a sentence to the caption of Suppl Fig. 1c, saying that “each particle is a 2D projection and alone does not provide adequate resolution for a high-resolution 3D structure”. (p. 24, lines 7-8)

- Page 5, line 1-2: could authors explain why Cd was used? Why not use Zn instead? This could also be stated in the results section.

Response: From previous crystal structure work, the protein has been observed to be more stable with Cd than Zn. We have revised the text to indicate this by adding the following text: “Cd²⁺ has been shown to stabilize the protein for crystallization”. We also cited the previous paper using Cd. (p. 5, lines 9-10)

- Page 5, line 21-22: What does that mean functionally? Can the site have its Cd replaced by Zn, although that does not happen in the crystal? Please make it clearer to the average reader.

Response: In the crystal structure of *BbZIP*, the Cd in the M2 site cannot be replaced by Zn. Functionally, this result suggests that the M2 site may be not essential for Zn transport. Indeed, site-directed mutagenesis for the M2 site indicates that the M2 site is not essential for Zn²⁺ transport. So, the observation of a partial occupancy of Cd²⁺ in the M2 site in the cryo-EM structure is consistent with a less essential role of the M2 site in the Zn²⁺ transport activity. We have clarified this point in the revised manuscript. (p. 5, lines 31-32; p. 6, lines 1-2). Due to a concern that Zn-bound *BbZIP* might be not stable enough for cryo-EM analysis, we haven't tried to replace Cd by Zn.

- Page 5, line 22-23: the phrase needs rewriting (please check it), although the meaning is understandable. However, I would suggest that authors to provide alignments to support the claim, instead of only citing a reference.

Response: We thank the reviewer for the suggestion and have revised the sentence to make it clear that the M1 site plays a major role. (p. 6, lines 3-4) And, as requested, we have added alignments (Suppl Fig. 7) to support the claim.

- Page 6, Line 4: please provide an alignment to support the claim of Glu276 conservation.

Response: We added an alignment figure (Supp Fig. 7) to show the conservation of Glu276.

- Page 6, line 13: “consistent”- please correct.

Response: Corrected.

- General comments on the model: the idea that the cytoplasmic, histidine-containing loop, works as an inhibitory module to avoid Zn toxicity is interesting, and the experimental data is consistent with that. However, it feels like other explanations are possible, since changes in movement and/or positioning of the loop under different Zn concentrations is not shown in the current paper. Molecular dynamics could help with that, and may be necessary to actually demonstrate the validity of the model.

Response: We thank the reviewer for the suggestion regarding the use of MD simulation. The main goal of Figure 4 is to summarize the structural and functional data in a plausible model. It's true that other explanations are possible. However, in our view, MD simulation may help

to explain our observed experimental structure and data if it works well. Having said this, MD simulation alone would be insufficient to validate the model. So, we plan to conduct follow-up cryo-EM and computational studies to investigate the conformational changes of the wild-type and His149/His151 mutants in the Zn-sensing loop in the presence of different concentrations of Zn. We have revised the text to acknowledge that alternative explanations are also possible. (p. 8, lines 12-14).

- Page 8, line 12-13:

Response: We didn't see any specific comment here. We nevertheless revised the sentence for improved clarity. (p. 8, lines 27-28)

- Page 8, line 14-16: this statement agrees with the idea that molecular dynamics might be needed to investigate the model and provide more evidence for its validity.

Response: We agree with the reviewer's suggestion. We will perform MD simulations to investigate the model in our follow-up studies to this work.

- Page 8, line 18-19: this claim is still a possibility. I would de-emphasize it.

Response: We have de-emphasized the claim in the following revised text "some living organisms may have evolved a mechanism to sense their intracellular Zn²⁺ status". (p. 8, lines 2-3)

Reviewer #2 (Remarks to the Author):

Zinc is an essential micronutrient that supports all living organisms through regulating numerous biological processes. However, the function and mechanism of uptake regulation by intracellular Zn²⁺ status are not fully understood. In this manuscript, Pang et al. reported the first cryo-electron microscopy structure of bacterial ZIP transporter from *Bordetella bronchiseptica* (BbZIP) at 3.06 Å. The obtained cryo-EM structure revealed that BbZIP adopted a dimeric configuration with each protomer in an inward facing, inhibited conformation. More importantly, a metal ion egress site was observed to be covered by the loop L(3,4) in the reported cryo-EM structure. A novel mechanism of autoregulating zinc uptake through a built-in sensor has also been proposed and validated biochemically.

Response: We thank the reviewer for this concise summary of our work.

Points:

1. P3, L31-32 "To enhance the signal to noise ratio of BbZIP images, we used a physical pixel size of 0.666 Å/pixel in super-resolution mode".

The final reconstructed map was resolved at 3.06Å resolution which is far from the Nyquist (1.32 Å). Authors should explain why collecting images at a much higher magnification perhaps be beneficial to small membrane protein. Otherwise, it would be misleading to the general cryo-EM practitioners.

Response: As pointed by the reviewer, the final reconstruction is at 3.06 Å which is substantially lower than the Nyquist resolution at 1.32 Å. For small membrane proteins of ~60 kDa and particle sizes of 60-70 Å, we reasoned that collecting data at 0.666 Å/pixel would give us sufficient numbers of pixels to calculate the background signals as well as their contrast from particle signals. In addition, a large number of pixels help provide a more accurate estimation of both signal and noise. Therefore, we used a small pixel size (0.666Å/pixel) to obtain an improved signal-to-noise ratio. We have added a sentence in the text to emphasize the purpose of a more accurate estimation of signal over noise from using a high magnification, i.e. a small pixel size. (p. 4, lines 4-7)

2. P4, L8-12 “Due to the presence of amphipols around BbZIP, standard ab initio 3D reconstruction in CryoSPARC 3.2 did not produce reasonable structural models. To make use of the structural features of transmembrane helices in ab initio reconstruction, we used data with resolution cutoffs of between 7 and 10 Å to generate four ab initio models from 50,000 randomly selected particles in CryoSPARC.”

Authors should provide a full gallery of 2D class averages and label the classes selected for ab initio models. Please explain the rationale in more detail why resolution cutoffs (7-10 Å) are applied. Is there any possibility that additional conformational states might be missing due to such a restricted resolution cutoff?

Response: We have included a new Suppl Fig. 2 to show a full gallery of 2D class averages (150 classes) and indicated the classes (135 classes) selected for the generation of ab initio models.

We apologize that we didn't make it clear with respect to the description of the resolution cutoff for the ab initio models. The cutoff is the high-resolution cutoff, and we used all the low-resolution data.

As shown in our revised Suppl Fig. 3, for the generated ab initio models, we did not observe additional conformations.

Authors performed data processing interchanging between Relion and cryoSPARC. However, Suppl Fig.3 is over simplified and extremely confusing. Authors should provide a comprehensive data processing workflow and update it in Suppl Fig.3.

Response: We have provided a comprehensive data processing workflow in our updated Suppl Fig. 3, specifying each step using cryoSPARC or Relion.

3. P4, L23-27 “In the final reconstruction, most particles represent a wide-spread distribution in the Euler space with slightly side-view particles (Supplementary Fig. 4e). The reconstructed map uses a 2-fold symmetry (C2) which displays a higher resolution than a separate reconstruction with no symmetry (C1) at a resolution of 3.3 Å (Supplementary Fig. 4d).”

It appears that the orientation plot shown in Suppl Fig.4e was from Relion. However, the final reconstruction FSC curve is obtained from cryoSPARC according to the corresponding text in page 21, which is confusing. Authors should report which FSC curves are actually shown in Suppl Fig.4d, masked or unmasked? In addition, FSC curve should be plotted all the way to Nyquist frequency (1.32 Å) not stop at 3 Å.

Response: Both the orientation plot and FSC curves are based on data from cryoSPARC. To plot Suppl Fig 4e, we converted the format of cryoSPARC particle orientation to the Relion .bild format for visualization. For the final reconstruction, we didn't use the pixel size of 0.666 Å/pixel. Instead, we binned the pixels by 2 to 1.332 Å/pixel. As suggested by the reviewer, we have updated the FSC curves to include unmasked and masked curves to the Nyquist resolution at 2.664 Å. (Suppl. Fig. 4d)

4. P5, L1-2 “We used cadmium in our protein purification; we found six Cd²⁺ ions in the dimer.”

Authors should carefully examine the C1 reconstruction map to validate the number of Cd²⁺ ions within a dimer. It would be informative to validate indeed the two protomers are in identical conformation in C1 map. Therefore, symmetry expansion followed by 3D classification without alignment should be performed to identify potential intermediate states.

Response: We thank the reviewer for this suggestion. During our cryoSPARC heterogenous and Relion nonalignment classifications, we didn't apply C2 symmetry (Suppl Fig. 4). We have made this clear in the revised method in the following text: "No symmetry was used during the ab initio model generation and heterogenous refinements in cryoSPARC". (p. 22, lines 22-23). We applied C2 symmetry only during our last homogenous reconstruction and non-uniform refinement within cryoSPARC. We have paid attention in search of possible conformations for this small membrane protein including the Cd²⁺ binding sites in each protomer. However, we didn't observe additional intermediate states associated with either Cd²⁺ binding or the repositioning of loops L(3,4). We added the following sentence to the Results section to clarify this: "We checked the two independent protomers in the C1 reconstruction and we did not observe any structural differences." (p. 5, lines 1-2) We hypothesize that these intermediate states may be not long-lived and/or may have been lost during sample vitrification. It's also possible that the number of particles belonging to intermediate states is insufficient to generate a high-resolution 3D structure. More micrographs and particles and more importantly new biochemistry may be needed to create and capture other states for single-particle cryo-EM. This is exactly the type of follow-on study we are planning to perform.

5. P5, L12-16 "The third ion binds to a site (M3) near the cytoplasmic surface (Fig. 2a) that consisted by residues Asp144 on TM3, Glu276 TM7, and two histidine residues, His149 and His151, on a loop (residues 146-166) between TM3 and TM4 (Fig. 2c). We thus named this loop as L(3,4). The loop has alternative conformations: disordered in the crystal structure and ordered in the cryo-EM structure (Supplementary Fig. 6)."

Solving the loop L(3,4) structure is the main structural discovery in this work. Loop L(3,4) should be highlighted in the local resolution map indicating the feasibility for building an atomic model particularly at His149 and His151.

Response: We have added a cryo-EM density covering the Loop L(3,4) including His149 and His151 to show the quality of the map for building an atomic model (please see the additional panel added as Suppl Fig. 6d). The cryo-EM density for His149 and His151 are now also shown in Fig. 2c.

6. P7, L14-31 "Mechanism of negative regulation of Zn²⁺ uptake".

The focus and main findings in the current manuscript solely support a potential mechanism that Zn²⁺ uptake could be negatively regulated at the inhibitory M3 site.

It lacks sufficient evidence with vast gap knowledge to explain the mechanism of BbZIP Zn²⁺ uptake through multiple hypothetical intermediate states during an entire cycle. It has been reported that elevator domain undergoes significant conformational changes as a rigid-body relative to scaffold domain upon substrate binding. The illustrated inward open state in Fig.4b is perhaps over simplified. Authors should revise Fig.4b in the updated manuscript.

Response: We thank the reviewer for bringing up the elevator model and conformational changes proposed in the recent Sci. Adv. Paper (DOI: 10.1126/sciadv.abn433). As pointed out by the reviewer, there are multiple hypothetical intermediate steps between the outward- and inward-facing states. For example, there are at least two inward-facing conformations, M1 and M1' as shown in Figures 3a and 3b. These comments were helpful and motivated us to revise Figure 4 to show conformational changes in the elevator domain relative to the scaffold domain. "The access of Zn²⁺ to the M1-site triggers a conformational switch from outward facing to an inward-facing conformation with the Zn²⁺ remaining in the M1-site as shown in Fig. 3a." We also revised Fig. 4a by adding Q207 for the M1 site and Fig. 4c by adding H177, E181, Q207 for the blocked M1 site.

7. P13, L27 "Data availability"

Author should deposit C1, C2 maps and the sharpened map for modeling in the EMDB. Authors should deposit meta data to EMPIRE upon publication.

Response: We have deposited both original and sharpened maps to EMDB. We have also deposited atomic coordinates and metadata to PDB.

8. P21, L13-16 “A total of 1,197,192 particles were picked, extracted at 256 pixels, and binned to 64 pixels with a pixel size of 2.664 Å. We used cryoSPARC for particle clean-up through 2D class averaging which resulted in 1,901,053 selected particles.”

Picked number of particles is less than the number of particles after cleaning up. It doesn't make sense.

Response: We thank the reviewer for pointing out our mistake. It should have been 1,997,192 as shown in Suppl Fig 3. It is now correctly stated in the revised manuscript.

In summary, Pang et al. solved the first cryo-EM structure of BbZIP transporter in dimeric conformation. The structural findings together with the biochemistry data support a novel mechanism of negative regulation of Zn²⁺ uptake through a built-in sensor. The obtained cryo-EM map looks impressive, however the structural findings behind are less exciting. The inward facing inhibited state structure is likely more homogenous and resolved from only a subset of particles. It is worth to investigate the full dataset more carefully, perhaps identify intermediate states with each protomer as the mechanistic interpretation could be drastically changed.

Response: We agree with the reviewer and hope to resolve additional conformational states in the *BbZIP* transport cycle in our follow-on studies. However, we think that new biochemistry will be required to obtain and stabilize additional states that are suitable for cryo-EM studies. In this regard, we initially plan to remove the N-terminal α -helical paddle and use a lower pH to explore the possibility of capturing additional conformational states.

Reviewers' Comments:

Reviewer #2:

Remarks to the Author:

I am mostly satisfied by author's responses. In addition to EM density map and coordinates, authors should deposit raw micrographs to EMPIAR. It is essential to examine the data processing reproducibility by the scientists in the field.